Novel application of the ferroptosis-related genes risk model associated with disulfidptosis in hepatocellular carcinoma prognosis and immune infiltration

Wei Jiayan 1
Wang Jinsong 1
Chen Xinyi 1
Zhang Li 2 syszhang@hotmail.com
Peng Min 1 mpeng320@whu.edu.cn
1 Department of Oncology, Renmin Hospital of Wuhan University , Wuhan, Hubei , China
2 Basic Medical Sciences, Wuhan University School of Basic Medical Sciences , Wuhan, Hubei , China
Zhang Xin
Electronic publication date: 2024 Feb 2
Publication date: 2024
Volume: 12
Electronic Location ID: e16819
Received 2023 Jul 13; Accepted 2023 Dec 31
Copyright: © 2024 Wei et al.
Copyright year: 2024
Copyright holder: Wei et al.
License: This is an open access article distributed under the terms of the Creative Commons Attribution License, which permits unrestricted use, distribution, reproduction and adaptation in any medium and for any purpose provided that it is properly attributed. For attribution, the original author(s), title, publication source (PeerJ) and either DOI or URL of the article must be cited.
License URL: https://creativecommons.org/licenses/by/4.0/

Keywords: Ferroptosis, Disulfidptosis, HCC, Prognosis model, Tumor microenvironment

Funding: National Science Foundation of China 81770169 Hubei Science Foundation for Distinguished Young Scholars 2023AFA079 Interdisciplinary Innovative Talents Foundation from Renmin Hospital of Wuhan University JCRCFZ-2022-025 Beijing Science and Innovation Medical Development Foundation KC2021-JX-0186-18 China primary healthcare foundation cphcf-2022-183 China Zhongguancun Precision Medicine Science and Technology Foundation ZLXGBXKYXM-030-02 This work was supported by grants from the National Science Foundation of China (No. 81770169), The Hubei Science Foundation for Distinguished Young Scholars (2023AFA079), The Interdisciplinary Innovative Talents Foundation from Renmin Hospital of Wuhan University (JCRCFZ-2022-025), the Beijing Science and Innovation Medical Development Foundation (KC2021-JX-0186-18), the China primary healthcare foundation (cphcf-2022-183), and the China Zhongguancun Precision Medicine Science and Technology Foundation (ZLXGBXKYXM-030-02). The funders had no role in study design, data collection and analysis, decision to publish, or preparation of the manuscript.

==============================
Hepatocellular carcinoma (HCC) stands as the prevailing manifestation of primary liver cancer and continues to pose a formidable challenge to human well-being and longevity, owing to its elevated incidence and mortality rates. Nevertheless, the quest for reliable predictive biomarkers for HCC remains ongoing. Recent research has demonstrated a close correlation between ferroptosis and disulfidptosis, two cellular processes, and cancer prognosis, suggesting their potential as predictive factors for HCC. In this study, we employed a combination of bioinformatics algorithms and machine learning techniques, leveraging RNA sequencing data, mutation profiles, and clinical data from HCC samples in The Cancer Genome Atlas (TCGA), Gene Expression Omnibus (GEO), and the International Cancer Genome Consortium (ICGC) databases, to develop a risk prognosis model based on genes associated with ferroptosis and disulfidptosis. We conducted an unsupervised clustering analysis, calculating a risk score (RS) to predict the prognosis of HCC using these genes. Clustering analysis revealed two distinct HCC clusters, each characterized by significantly different prognostic and immune features. The median RS stratified HCC samples in the TCGA, GEO, and ICGC cohorts into high-and low-risk groups. Importantly, RS emerged as an independent prognostic factor in all three cohorts, with the high-risk group demonstrating poorer prognosis and a more active immunosuppressive microenvironment. Additionally, the high-risk group exhibited higher expression levels of tumor mutation burden (TMB), immune checkpoints (ICs), and human leukocyte antigen (HLA), suggesting a heightened responsiveness to immunotherapy. A cancer stem cell infiltration analysis revealed a higher similarity between tumor cells and stem cells in the high-risk group. Furthermore, drug sensitivity analysis highlighted significant differences in response to antitumor drugs between the two risk groups. In summary, our risk prognostic model, constructed based on ferroptosis-related genes associated with disulfidptosis, effectively predicts HCC prognosis. These findings hold potential implications for patient stratification and clinical decision-making, offering valuable theoretical insights in this field.

Introduction

Liver cancer is a significant global contributor to cancer-related mortality, and its incidence is anticipated to rise considerably by 2025 (Llovet et al., 2021; Ganesan & Kulik, 2023). HCC predominates as the primary liver cancer, comprising 90% of cases, making it the most prevalent pathological type (Forner, Reig & Bruix, 2018). Prominent risk factors for HCC encompass cirrhosis, hepatitis B and C infections, and alcohol abuse (Anwanwan et al., 2020). For early-stage HCC patients, surgical resection stands as the preferred therapeutic approach, though the 5-year recurrence rate is alarmingly high, reaching 70% (Liu et al., 2021). The last decade has witnessed a transformative wave in the realm of advanced HCC treatment, marked by the introduction of molecularly targeted drugs (e.g., kinase inhibitors), angiogenesis inhibitors (e.g., bevacizumab), and immune checkpoint inhibitors (e.g., nivolumab), which have kindled fresh hope for patients (Lee, Guan & Ma, 2022). Nevertheless, the benefit of these treatments is constrained to a limited patient pool, and the specter of drug resistance, recurrence, and metastasis looms large (Oura et al., 2021; Tang et al., 2022). To recap, the prognosis for HCC remains bleak. Hence, there is an urgent and imperative need to explore novel approaches for screening, diagnosis, and therapy to enhance HCC prognosis. The development of a risk prognosis model holds the potential to stratify patients, conferring substantial importance in the realms of survival prediction and clinical decision-making.

In early 2023, a novel mode of cell demise, termed disulfidptosis, was unveiled, setting it apart from conventional forms like apoptosis, cuproptosis, and ferroptosis. Disulfidptosis is characterized by an anomalous collapse of disulfide bonds within the actin cytoskeleton, stemming from an accumulation of cystine within the cell (Liu et al., 2023). The starvation of glucose in cells with heightened SLC7A11 expression can incite disulfide-induced cell death. Furthermore, researchers noted that glucose transporter inhibitors triggered disulfidptosis in tumor cells with elevated SLC7A11 expression, consequently impeding tumor growth (Liu et al., 2023). Numerous genes connected to disulfidptosis have been pinpointed, including glycogen synthetase (GYS1) and various genes involved in mitochondrial oxidative phosphorylation (e.g., NDUFS1, NDUFA11, NUBPL, and LRPPRC), whose deactivation synergizes with glucose deprivation to provoke cell disulfidptosis (Liu et al., 2023). This discovery of a novel mechanism wherein disulfide bonds precipitate cytoskeletal collapse unveils potential targets for modulating this mode of cell death and devising innovative cancer treatment strategies (Machesky, 2023).

Like disulfidptosis, ferroptosis represents an alternative mode of cell demise, characterized by its iron-dependent nature, and it plays a pivotal role in tumorigenesis and tumor progression. Throughout the evolution of tumors, ferroptosis serves a dual purpose, both promoting and restraining tumor growth (Huang et al., 2023). Furthermore, it exerts a substantial impact on the efficacy of chemotherapy, radiotherapy, and immunotherapy in cancer patients (Yuan et al., 2021; Wang et al., 2019). Notably, it has been observed that the metabolic reconfiguration of tumors is intrinsically tied to their sensitivity to ferroptosis. In particular, tumor cells exhibiting heightened metabolic activity display a heightened susceptibility to ferroptosis (Chen et al., 2021; Friedmann, Krysko & Conrad, 2019). Consequently, manipulating tumor metabolism to trigger ferroptosis emerges as a promising therapeutic avenue for inducing cancer cell demise, particularly in cases where traditional treatments falter. Several signaling pathways in the liver oversee the orchestration of ferroptosis (Chen et al., 2022). In the context of advanced liver cancer therapy, the molecularly targeted drug Sorafenib has exhibited anti-tumor effects linked to ferroptosis (Li et al., 2021). ATF4 and YAP/TAZ, however, can foster resistance to Sorafenib in liver cancer by impeding ferroptosis (Gao et al., 2021). Conversely, targeting the ferroptosis inhibitor NRF2/GPX4 axis can bolster Sorafenib-induced cell death, thereby improving the efficacy of drug therapy (Wang et al., 2021). Intriguingly, research has unveiled that inducing ferroptosis in mice does not inhibit the onset of HCC but rather elicits immunosuppression (Conche et al., 2023). In conclusion, the specific regulatory effect of ferroptosis on the occurrence and progression of HCC remains inconclusive, and the specific mechanism involved remains to be fully elucidated.

Owing to the limited number of disulfidptosis-related genes (DRGs), we performed a correlation analysis between ferroptosis-related genes (FRGs) and DRGs to identify FRGs associated with disulfidptosis. This investigation aimed to elucidate the connection between ferroptosis, disulfidptosis, and the prognosis of HCC. Consequently, we integrated multiple DRGs-FRGs by scrutinizing publicly available sequencing datasets, thereby constructing a risk prognostic model for HCC. Furthermore, we assessed the drug sensitivity of this model to establish a theoretical foundation for the stratified clinical management of HCC patients. Figure 1 depicts the flowchart of this study.

Figure 1 Data acquisition and processing flowchart.

Materials and Methods

Data acquisition

We retrieved clinical information, survival data, gene mutation details, and RNA sequencing data (RNA-seq) from The Cancer Genome Atlas (TCGA) database, encompassing a total of 415 samples, comprising 365 HCC samples and 50 normal samples. Subsequently, we acquired validation cohorts for HCC, GSE14520, from the Gene Expression Omnibus (GEO) database, and ICGC data from the International Cancer Genome Consortium (ICGC) database. The GSE14520 cohort included 242 HCC samples, and the ICGC cohort included 260 HCC samples, none of which contained normal liver tissue samples. To obtain 259 FRGs, we accessed the FerrDb website (http://www.zhounan.org/ferrdb/legacy/operations/download.html). We sourced 10 DRGs from a prior study (Liu et al., 2023). Subsequently, a Spearman correlation analysis between DRGs and FRGs was carried out, and FRGs with nonzero correlation coefficients and p < 0.05 were considered to be associated with DRGs. Finally, 174 DRG-FRGs were picked out.

Analysis of differentially expressed genes

We aligned RNA-seq data of HCC samples from the TCGA cohort with the set of 174 DRG-FRGs. We employed the “limma” package in the R software to analyze the differentially expressed genes (DEGs) in the RNA-seq data from both HCC tissue and non-tumor tissue samples. |log fold change (FC)| > 1, the gene for adjusted p < 0.05 was screened out to obtain DE-DRG-FRGs (Sun et al., 2022). Log FC > 1 represents genes upregulated in tumor tissues, while Log FC < -1 represents genes downregulated. Heatmap and volcano plot were generated to show the DE-DRG-FRGs. For univariate Cox regression analysis, we utilized the R “forestplot” package and identified DE-DRG-FRGs with a p-value less than 0.01, which were considered independent prognostic factors associated with overall survival (OS) in HCC (He, Yang & Jin, 2022). To explore the protein-protein interactions (PPI) among DE-DRG-FRGs associated with OS, we constructed a network using the STRING website (http://string.embl.de/) (Wang et al., 2021).

Enrichment analysis of DEGs

We conducted KEGG (Kyoto Encyclopedia of Genes and Genomes) and GO (Gene Ontology) enrichment analyses to elucidate the biological processes and signaling pathways enriched in the DE-DRG-FRGs using the R “clusterProfiler” package (Lian et al., 2021). To investigate the potential associations of DE-DRG-FRGs with liver disease, we performed Disease Ontology (DO) analysis using the “DOSE” software package (Wang et al., 2023).

Unsupervised cluster analysis

We employed the R software package “ConsensusClusterPlus” for unsupervised clustering analysis, utilizing OS-associated DE-DRG-FRGs as the basis (Zeng et al., 2022). To assess cluster differentiation, we utilized Principal Component Analysis (PCA), Uniform Manifold Approximation and Projection (UMAP), and t-Distributed Stochastic Neighbor Embedding (t-SNE). Survival analysis of different clusters in HCC was conducted using the R “survival and survminer” package. HCC samples from the GSE14520 and ICGC cohorts were also performed unsupervised clustering and survival analysis as validation cohorts.

Analysis of biological behavior and tumor microenvironment (TME) on phenotypes of DRG-FRGs

“c5.go.symbols” and “c2.cp.kegg.symbols” gene sets were downloaded, and the R software “GSVA” package was used to perform the Gene Set Variation Analysis (GSVA) to explore the differences in molecular function, biological behavior, and signaling pathways of different phenotypes based on DRG-FRGs. Online website TIMER 2.0 (http://timer.cistrome.org/) including diverse algorithms (Jiang et al., 2021) of TIMER, CIBERSORT, CIBERSORT-ABS, QUANTISEQ, MCPCOUNTER, XCELL, and EPIC was used to conduct analysis of immune cell infiltration, and a heatmap to display the differences. Single-sample gene set enrichment analysis (ssGSEA) plays a role in calculating the enrichment score that indicates the level at a gene is enriched in a particular gene set. ssGSEA was performed using R software packages “GSVA” and “GSEABase” to calculate the concentration score of 23 immune cells in different clusters (Xiong et al., 2023). The ESTIMATE algorithm is a method of calculating tumor purity (Jiang et al., 2021). The R software “ESTIMATE” package was used to analyze the immune scores and stromal scores in different clusters to explore the infiltration of immune cells and stromal cells in TME. Furthermore, the infiltration data of immune cell in HCC were obtained from the online website CIBERSORT (Liu et al., 2023) (https://cibersortx.stanford.edu/), and the R “ggpubr” and “limma” packages (Dong et al., 2023) were used to visualize the differences in clusters.

Construction and validation of a risk prognostic model based on DRG-FRGs

The TCGA cohort was used as the training cohort, and GEO and ICGC cohorts as the validation cohorts. We used DE-DRG-FRGs to construct a risk prognostic model to predict immune features, survival time, and drug sensitivity in HCC patients. First, we have previously performed univariate Cox regression analysis to screen out DE-DRG-FRG associated with OS as model candidate genes. Subsequently, to avoid collinearity and overfitting, the least absolute shrinkage and selection operator (LASSO) were conducted with the R “glmnet” package (Xu et al., 2023) to complete the compression of the coefficients and eliminate genes with zero coefficient (Lin et al., 2023). Finally, we chose the genes with non-zero coefficients to carry on multivariate Cox regression analysis, and the regression coefficients of each gene were obtained. The risk score was a linear combination of multivariate Cox regression coefficients and gene expression levels. The calculation formula of RS was as follows, RS = regression coefficient 1 * gene 1 expression + regression coefficient 2 * gene 2 expression + … + regression coefficient n * gene n expression. According to this formula, the RS of HCC samples in the TCGA cohort and the validation cohort GSE14520 and ICGC were calculated, respectively. Based on the median RS, the HCC samples in the TCGA, GEO and ICGC cohort were all divided into high-and low-risk groups. In R software, Kaplan–Meier analysis was performed using the “survminer” package to compare survival differences in the high-and low-risk groups across the three cohorts (Liu et al., 2021). The accuracy of the risk model in predicting survival was evaluated with the receiver operating characteristic (ROC) curves plotted using the R software “timeROC” package (Song et al., 2022). Risk plots were mapped with the R “pheatmap” package (Zhang et al., 2023) to evaluate health status, risk score distribution, and expression of model genes in the two risk groups. In addition, gene correlations were visualized using the R “corrplot” package (Zhang et al., 2023), and the Kaplan–Meier survival analysis on six signature genes were performed. RS differences for different clinical features were plotted with the R “ggpubr” and “limma” packages, and survival curves for two risk groups under different clinical features were plotted. Based on clinical features including age, gender, pathological grade, T (tumor size), stage and RS, univariate and multivariate Cox regression analysis were carried out. Finally, nomogram model with a corresponding calibration curve were conducted with the R “rms” package (Zou et al., 2023) to predict HCC survival by combining multiple clinical features and RS.

Decision curve analysis and model comparison

The R “timeROC” package was used to draw ROC curves to assess the predictive efficacy of the nomogram model. R “ggDCA” package was used to conduct decision curve analysis (DCA) to measure the net clinical benefit of the nomogram model (Clift et al., 2023). The ROC and survival curves of multiple HCC prognostic models were plotted with the R software “timeROC” and “survminer” packages, respectively.

Analysis of gene mutation and tumor mutation burden

Somatic mutations of HCC were downloaded from cBioPortal (http://www.cbioportal.org/) database and R “maftools” package was used to analyze the incidence of somatic mutations in the two risk groups (Wang et al., 2023). Tumor mutation burden (TMB) is referred as somatic mutations numbers in the tumor genome exon coding region after germline mutations are removed. The “TMB” function in the R software “maftool” package was used to calculate the TMB of high-and low-risk groups, subsequently the TMB differences between the two groups were analyzed. Finally, the R “survminer” package was used to draw the survival curves of patients with different TMB levels.

Gene set enrichment analysis (GSEA)

The R “org. Hs. eg. db” package was used in the GSEA to explore biological behavior and signaling pathways (Wang et al., 2022). The c5.go.v7.5.1.symbols and c2.cp.kegg.v7.5.1.symbols gene sets were downloaded from the Molecular Signatures Database (MSigDB) (Abdel-Aziz et al., 2023) (https://www.gsea-msigdb.org/gsea/msigdb/).

Analysis of tumor microenvironment

The analysis on the infiltration features of 23 immune cells in the two risk groups was also performed with ssGSEA. Also on the TIMER 2.0 website, the infiltration features of immune cells were analyzed and visualized with a heatmap. The scatterplots displaying the association between model genes and immune cell infiltration were downloaded using the online website TIMER (Peng et al., 2022). Similarly, R software “ggpubr” and “limma” packages were used to visualize the difference of immune cell infiltration between the two risk groups.

Analysis of response to drug therapy

The Genomics of Drug Sensitivity in Cancer (GDSC) (https://www.cancerrxgene.org/) is a website to obtain the drug sensitivity information files. The R software “pRRophetic” package (Ding et al., 2023) was used to calculate the half maximal inhibitory concentration (IC50) to analyze the predictive ability of the risk prognostic model for hundreds of antitumor drugs sensitivity. IC50 is used as an evaluation index of drug sensitivity, which was negatively correlated with drug sensitivity (Wang et al., 2020). IC50 differences between the two risk groups were visualized with R software “ggplot2” package.

Analysis of cancer stem cell infiltration

Cancer stem cells are subsets of cells with stem cell-like characteristics that play an important role in tumor proliferation, metastasis, and recurrence. In the TCGA cohort, DNA methylation-based stemness score (DNAss) and RNA-based stemness score (RNAss) were acquired from the UCSC Xena browser (Wang et al., 2022) (http://xena.ucsc.edu/) website. Subsequently, the correlation of DNAss, RNAss and risk score was analyzed.

Single-cell analysis

The online website TISCH (Yao et al., 2023) (http://tisch.comp-genomics.org) was used to analyze the expression of model genes in single cells of HCC tissues.

Analysis of immune checkpoints and antigen presentation

The immune checkpoint (IC) is an inhibitory pathway of the immune system, which is closely related to human immune status. HLA is the expression product of the human major histocompatibility complex (MHC), which is found on the surface of many immune cells and contributes to antigen presentation. The R software “limma” package was used to analyze the difference between ICs and HLA expression in high-and low-risk groups.

Validation of signature genes in databases

The protein staining results of 6 DRG-FRGs in liver cancer tissues and liver normal tissues were obtained the Human Protein Atlas (HPA) (Chen et al., 2023) (https://www.proteinatlas.org/) database. The mRNA expression of genes in different cell lines of HCC was analyzed in the Cancer Cell Line Encyclopedia (CCLE) (http://www.broadinstitute.org/ccle) database (He et al., 2023).

Immunohistochemistry (IHC)

Tissue paraffin sections were placed in xylene and heated for 15 min for dewaxing. Hydration treatment was used with different concentrations of alcohol (absolute ethanol, 85%, 75%). Then sections were put into EDTA antigen retrieval buffer (PH9.0) and microwave for 23 min. After natural cooling, it was placed in PBS and washed three times on a decolorization shaker for 5 min each. Add primary antibody (1:400~1:100, different antibody ratios are different) and incubate overnight in a humidified box at 4 °C. In the secondary day, PBS was used to wash off the primary antibody, and the secondary antibody was added. Subsequently, these sections were incubated at a room temperature for 50 min. The secondary antibody was washed off with PBS. The sections were developed with DAB. After completing color development, the nucleus was stained with hematoxylin.

Statistical analysis

Overall statistical analysis and charts making were completed on R software version 4.2.1 (R Core Team, 2022). The differences between the two groups were compared using the Wilcoxon rank test. The Spearman test was used to test the correlation between two variables. Heatmaps, volcano maps, and violin plots were drawn with the R software “pheatmap”, “ggplot2” and “ggpubr” packages, respectively. The Kaplan-Meier curves and log rank test were carried out to analyze the OS differences between groups.

Results

Identification and analysis of differentially expressed ferroptosis genes associated with disulfidptosis

Differentially analysis showed that there were 72 DE-DRG-FRGs in HCC tissues and non-tumor tissues (Fig. 2A). Among these, 71 genes exhibited up-regulation, while only one gene demonstrated down-regulation in the tumor tissue (Fig. 2B). GO enrichment analysis showed that DE-DRG-FRGs were mainly enriched in the biological activities of response to chemical and oxidative stress inside and outside the cell (Fig. 2C). KEGG enrichment analysis highlighted the primary enrichment of DEGs in signaling pathways involving autophagy, mTOR, and hepatocellular carcinoma, among others (Fig. 2D). DO analysis uncovered associations of DE-DRG-FRGs with various malignancies, including those of the nervous system and the female reproductive system (Fig. 2E). Univariate Cox regression analysis indicated that 16 out of the 72 DE-DRG-FRGs were independently linked to OS in patients with HCC (Fig. 2F). Among these, the expression of 10 genes exhibited a positive correlation with OS, while six genes displayed a negative correlation with OS (Figs. S1A–S1P). The PPI network illustrated the correlation of 16 OS-associated DE-DRG-FRGs and their respective contribution to prognosis (Fig. 2G). In summary, our study identified 72 DEGs between HCC tissues and non-tumor tissues, of which 16 were associated with OS.

Figure 2 Differentially expressed genes and functional enrichment analysis.

(A) Heatmap of 72 DE-DRG-FRGs. (B) Volcano plot of 72 DE-DRG-FRGs. (C) GO enrichment analysis of 72 DE-DRG-FRGs. (D) KEGG enrichment analysis of 72 DE-DRG-FRGs. (E) DO enrichment analysis of 72 DE-DRG-FRGs. (F) Univariate Cox regression analysis of 16 OS-related DE-DRG-FRGs. (G) PPI network of 16 OS-related DE-DRG-FRGs. |log FC| > 1, adjusted p < 0.05 was screened out to obtain DE-DRG-FRGs. DE-DRG-FRGs, differentially expressed ferroptosis-related genes associated with disulfidptosis-related genes; OS, overall survival; PPI, protein-protein interactions.

Unsupervised cluster analysis

Utilizing the 16 OS-related DRG-FRGs, we conducted unsupervised clustering analysis. This analysis identified K = 2 as the optimal number of clusters, classifying HCC patients in the TCGA cohort into two distinct clusters, denoted as Cluster A and B (Figs. 3A and 3B). Subsequent verification through PCA, t-SNE, and UMAP analyses confirmed a substantial differentiation between these two clusters (Fig. 3C, Figs. S2A and S2B). Survival analysis indicated that Cluster B exhibited improved OS benefits compared to Cluster A (Fig. 3D). We also conducted cluster analyses in the GSE14520 and ICGC validation cohorts (Figs. S2C, S2E, S2F–S2H). To elucidate the underlying mechanisms responsible for the survival disparity between the two clusters, we investigated variations in immune features. Heatmap analysis employing multiple algorithms revealed higher immune cell infiltration scores in Cluster A (Fig. 3H). ssGSEA further confirmed that Cluster A exhibited a superior immune status compared to Cluster B (Fig. 3I). CIBERSORT analysis highlighted substantial differences in immune cell infiltration features between the two clusters (Fig. S2I). Cluster A displayed significantly higher immune scores and ESTIMATE scores, with no significant difference in stromal scores (Fig. 3E). Cluster A had lower tumor purity, potentially accounting for the shorter survival time observed (Zhang et al., 2017). In addition, the GO GSVA result showed that Cluster B was significantly enriched in molecular biological activities such as lipid metabolism, amino acid metabolism, and iron ion binding (Fig. 3F). KEGG GSVA result showed that Cluster B was significantly enriched in signaling pathways related to amino acid and lipid metabolism (Fig. 3G). In summary, our unsupervised clustering analysis identified two distinct clusters in HCC, characterized by significant differences in both prognosis and the tumor immune microenvironment.

Figure 3 HCC clusters based on DRG-FRGs in the TCGA cohort.

(A) Unsupervised clustering based on 16 OS-related DRG-FRGs in 365 HCC samples in the TCGA cohort and optimal consensus matrices for k = 2. (B) Cumulative distribution function (CDF) for k = 2–9. (C) Principal component analysis (PCA) of clusters A (106 samples) and B (259 samples). (D) Survival analysis of clusters A and B. (E) ESTIMATE analysis of clusters A and B. (F) GO GSVA analysis of clusters A and B. (G) KEGG GSVA analysis of clusters A and B. (H) Heatmap of immune cell infiltration in clusters A and B. (I) Single-sample gene set enrichment analysis (ssGSEA) of clusters A and B. DRG-FRGs, ferroptosis-related genes associated with disulfidptosis-related genes; OS, overall survival; GSVA, gene set variation analysis. *p < 0.05; **p < 0.01; ***p < 0.001.

Construction and validation of risk prognostic model based on DRG-FRGs

Initially, a univariate Cox regression analysis identified 16 DRG-FRGs associated with OS as candidate genes. Subsequently, employing penalty and compression coefficients, 11 out of the initial 16 candidate genes were selected (Figs. 4A and 4B). Finally, multivariate Cox regression analysis was performed, and 6 DRG-FRGs (FTH1, G6PD, PML, SLC1A5, SLC7A11, STMN1) were included in our risk prognostic model (Fig. 4C). The risk score was calculated using the following formula, RS= (−0.178977181838147)*FTH1 + 0.203421627375275*G6PD + (−0.537653732806195)*PML + 0.14649163633218*SLC1A5 + 0.145028457083383*SLC7A11 + 0.190272995477894*STMN1. Risk scores were computed in two validation cohorts using the same formula. Within the three cohorts, samples were stratified into high-and low-risk groups based on the median RS. Survival analysis demonstrated a substantially superior OS outcome in the TCGA cohort’s low-risk group compared to the high-risk group (p < 0.001) (Fig. 4D). Corresponding results were observed in the other two validation cohorts, GSE14520 (p < 0.001) and ICGC (p = 0.004) (Figs. 4F and 4H). The area under the curves (AUC) for predicting survival time at 1, 3, and 5 years were 0.786, 0.704, and 0.683, respectively (Fig. 4E). These values indicate the high predictive accuracy of the risk model. The AUCs of ROC curves plotted for the GSE14520 and ICGC cohorts further demonstrated the favorable predictive power of our risk model for survival (Figs. 4G and 4I). In the low-risk group, higher expression levels of FTH1 and PML were observed, whereas in the high-risk group, higher expression levels of G6PD, SLC1A5, SLC7A11, and STMN1 were evident. Additionally, higher RS and a greater number of deaths were both concentrated in the high-risk groups (Fig. 4J). Figure 4K showed the correlation of the 6 DEG-FRGs. Univariate Cox regression analysis discovered that 6 DRG-FRGs were all independently related to OS (Fig. 4L). Specifically, the expression levels of FTH1 and PML were positively correlated with survival, whereas G6PD, SLC1A5, SLC7A11, and STMN1 displayed a negative correlation with survival (Figs. S3A–S3F). This was consistent with the distribution of DRG-FRGs in the two risk groups. In conclusion, the risk prognosis model effectively differentiated the prognoses of high-and low-risk groups and provided accurate predictions for the survival of HCC.

Figure 4 Identification of signature genes and prognosis analysis.

(A) LASSO coefficient profile plots of the 16 DRG-FRGs. (B) Regression algorithm using LASSO, 10 cross-validation method was used to select the optimal parameter (lambda). (C) Multivariate Cox regression analysis for six signature genes and corresponding regression coefficients. (D) The risk score, survival status, and gene expression distribution of HCC in the high-and low-risk groups in the TCGA cohort. (E) Correlation plot of six signature genes in the TCGA cohort. (F) Univariate Cox analysis showed that six signature genes were associated with OS. (G–I) Survival analysis of high-and low-risk groups in the TCGA, GEO, and ICGC cohorts. (J–L) Receiver operating characteristic (ROC) curves were used to analyze the predictive power of the prognostic model at 1, 3, and 5 years in the TCGA, GEO, and ICGC cohorts. DRG-FRGs, ferroptosis-related genes associated with disulfidptosis-related genes; OS, overall survival.

Evaluation of the risk prognostic model based on DRG-FRGs

Both univariate and multivariate Cox regression analyses both showed that RS was an independent prognostic factor in HCC (Figs. 5A and 5B). Soon afterwards, we successfully construct a nomogram model containing risk score and clinical features (age, sex, pathological grade, T, and stage) to predict the survival of HCC (Fig. 5C). The calibration plot illustrated a high degree of consistency between the predicted and observed survival (Fig. 5D). An analysis of risk score distributions within distinct clinical feature groups revealed significantly higher risk scores in cluster A, grade 3-4, T3-T4, and stage III-IV (p < 0.05), with no notable variation observed in terms of age and gender (Figs. S3G–S3L). The Kaplan-Meier curves revealed that RS was a powerful prognosis factor in HCC patients with different ages, genders, grades, T, and stages (Figs. 6A–6J). In addition, HCC patients with different clinical features also had better survival in the low-risk group. We subsequently assessed and compared the predictive capacity of our nomogram model. The area under the curve (AUC) values for 1-, 3-, and 5-year survival in ROC curves indicated that our nomogram model outperformed other clinical features (age, gender, T, and stage), with the RS following closely (Figs. 6K–6M). The 1-, 3- and 5- year DCA curves also indicated our nomogram model has superior ability to make clinical decisions (Figs. 6N–6P). To compare predictive ability of our model with other prognosis models in HCC, we also plotted ROC curves. AUCs for 1-, 3- and 5-year OS discovered that our risk prognostic model for HCC had better predictive ability than other models (Liu et al., 2019; Tian et al., 2021; Zhang et al., 2020) (Figs. 5E–5H). In addition, Kaplan–Meier curves showed that our risk model could most significantly distinguished survival differences in high-and low-risk groups (Figs. 5I–5L). Collectively, these findings indicate that our risk prognostic model possesses strong predictive capabilities for HCC survival.

Figure 5 Construction of the nomogram model and comparison of the prognostic models.

(A) Univariate Cox regression for RS and clinical features in the TCGA cohort. (B) Multivariate Cox regression for RS and clinical features in the TCGA cohort. (C) The nomogram for predicting 1-, 3-, and 5-year OS of HCC in the TCGA cohort. (D) The calibration curves for predicting 1-, 3-, and 5-year OS of HCC. (E–H) ROC curves for 1-, 3-, and 5-year OS prediction of HCC in different prognostic models. (I–L) Survival analysis of high-and low-risk groups in different prognostic models. RS, risk score; OS, overall survival; ROC, receiver operating characteristic.

Figure 6 Survival analysis of HCC with different clinical features and comparison of the predictive power of the nomogram model and other clinical features.

(A–J) Survival analysis of HCC with different clinical features in the TCGA cohort. (K-M) ROC curves for 1-, 3-, and 5-year OS prediction of nomogram model and other clinical features in the TCGA cohort. (N-P) Decision curve analysis (DCA) of the nomogram model and other clinical features in the TCGA cohort. OS, overall survival; ROC, receiver operating characteristic.

Analysis of gene mutation and tumor mutation burden

To assess the utility of the 6 DRG-FRGs in developing diagnostic, prognostic, and recurrence models, we investigated genetic alterations using the online cBioportal database. Our analysis revealed that 6.2% of patients exhibited mutations in these six genes, with G6PD displaying the most substantial mutation frequency, followed by FTH1, which featured notable amplifications and deep depletions (Fig. 7A). In the TCGA cohort, the high-risk group exhibited an 89.89% incidence of gene mutations, significantly surpassing the 81.46% observed in the low-risk group. Notably, TP53, CTNNB1, and TTN were the genes with the highest mutation frequencies in both risk groups (Figs. 7B and 7C). TMB serves as a promising biomarker for predicting the response to immune checkpoint inhibitors (ICIs). Tumors with elevated mutation burdens typically yield more neoantigens, rendering them more immunogenic and, consequently, more likely to respond favorably to ICIs. As illustrated in Fig. 7D, the TMB level was markedly higher in the high-risk group compared to the low-risk group (p < 0.05), suggesting that the high-risk group might exhibit a more robust response to ICIs. Survival analysis revealed that low TMB had longer survival time than high TMB (Fig. 7E). To explore the co-effect of risk score and TMB on prognosis of HCC patients, we divided patients into four groups for Kaplan-Meier survival analysis. The results revealed that patients with both low RS and low TMB had the longest survival time (Fig. 7F). In summary, the analysis of gene mutations and tumor mutation burden unveiled distinctions in gene mutations and potential variations in the tumorigenesis and progression of HCC between the high-risk and low-risk groups.

Figure 7 Gene mutation and tumor mutation burden (TMB) analysis of the prognostic model.

(A) Mutation analysis of six signature genes in the TCGA cohort. (B and C) Mutation landscape of high-and low-risk groups in the TCGA cohort. (D) Comparison of TMB in high-and low-risk groups in the TCGA cohort. (E) Survival analysis of high-and low-TMB in the TCGA cohort. (F) Survival analysis of TMB and risk score in the TCGA cohort.

Analysis of immune features and antigen presentation

To elucidate the mechanisms contributing to the divergent survival probabilities in the two HCC risk groups, we conducted an analysis of immune cell features. Analysis of immune cell features revealed a more pronounced infiltration of various immune cells in the high-risk group, including dendritic cells (DCs), macrophages, Th1 cells, and Th2 cells (Fig. 8A). CIBERSORT analysis demonstrated that the mRNA expression levels of memory B cells, activated CD4 memory T cells, Tregs, T cells gamma, T cells follicular helper, and M0 macrophages were notably elevated in the high-risk group compared to the low-risk group. In contrast, resting memory CD4 T cells, naive B cells, monocytes, and resting mast cells exhibited significantly higher expression levels in the low-risk group (Fig. 8B). The heatmap depicted marked disparities in the abundance of various infiltrated immune cells between the two risk groups, as determined by multiple algorithms (Fig. 8C). HLAs play a critical role in antigen presentation and are thus positively correlated with the efficacy of immunotherapy (Schaafsma et al., 2021). The high-risk group exhibited higher expression levels of both class I and class II HLAs, suggesting a potentially more favorable response to ICIs (Fig. 8D). ICIs are a pivotal approach to anti-tumor therapy, primarily functioning by inhibiting the interaction between ICs on the surface of tumor cells and IC receptors. The analysis demonstrated higher expression levels of ICs in the high-risk group, suggesting a greater likelihood of the effectiveness of ICIs in this group (Fig. 8E). These six signature genes were found to be correlated with the infiltration of immune cells. Specifically, FTH1 displayed positive associations with macrophage, dendritic cell infiltration, B cells, and neutrophils, but not with CD8+ T cells and CD4+ T cells (Fig. S4A). SLC7A11 expression exhibited positive correlations with neutrophil and dendritic cell infiltration, while no significant associations were observed with the other four types of immune cells (Fig. S4E). The expression of G6PD, PML, SLC1A5, and STMN1 were positively related to the infiltration level of six types of immune cells (Figs. S4B–S4D and S4F). In conclusion, these findings indicate disparities in the immune features of the TME between the high-and low-risk groups.

Figure 8 Immune infiltration analysis of the prognostic model.

(A and B) Infiltration score of immune cells in high-and low-risk groups in the TCGA cohort. (C) The heatmap of immune cell infiltration in high-and low-risk groups in the TCGA cohort based on multiple algorithms. (D) Expression of HLA in high- and low-risk groups in the TCGA cohort. (E) Expression of immune checkpoints in high-and low-risk groups in the TCGA cohort. HLA, human leukocyte antigen. *p < 0.05; **p < 0.01; ***p < 0.001

Gene set enrichment analysis

Figure 9 illustrates significantly enriched biological activities and signaling pathways. GO GSEA indicated that the high-risk group exhibited significant enrichment in molecular functions, including immunoglobulin production, leukocyte-mediated immunity, membrane invagination, production of molecular mediators of immune response, and immunoglobulin complex (Fig. 9A). Conversely, the low-risk group primarily displayed enrichment in processes such as alpha amino acid catabolism, cellular amino acid catabolism, lipid oxidation, monocarboxylic acid catabolism, and monooxygenase activity (Fig. 9B). KEGG GSEA identified the high-risk group as predominantly enriched in signaling pathways related to the cell cycle, the interaction between cytokines and cytokine receptors, DNA replication, neuroactive ligand-receptor interaction, and ribosome (Fig. 9C). In contrast, the low-risk group was primarily enriched in pathways associated with drug metabolism cytochrome P450, complement and coagulation cascades, fatty acid metabolism, peroxisome, and retinol metabolism (Fig. 9D). In summary, the primary enriched biological processes and signaling pathways in HCC exhibited significant differences between the high-and low-risk groups.

Figure 9 Gene set enrichment analysis (GSEA) of the prognostic model.

(A and B) GO enrichment analysis in high-and low-risk groups in the TCGA cohort. (C and D) KEGG enrichment analysis in high-and low-risk groups in the TCGA cohort.

Drug sensitivity analysis

We conducted an analysis of IC50 values, revealing a significant difference in IC50 between the two risk groups. IC50 values for drugs including 5-Fluorouracil, docetaxel, and paclitaxel were notably elevated in the low-risk group, whereas drugs like Sorafenib, Axitinib, and Oxaliplatin exhibited higher IC50 values in the high-risk group (Figs. 10A–10P). In summary, a noteworthy disparity in the sensitivity of HCC to antitumor drugs is evident between the high-and low-risk groups.

Figure 10 Drug sensitivity analysis of the prognostic model.

(A–P) Analysis of different drugs in high- and low-risk in the TCGA cohort. Due to the large variety of drugs, only partial charts of drugs were shown.

Analysis of cancer stem cell infiltration

We examined the variance in tumor stemness characteristics between the two risk groups using DNAss and RNAss. Our analysis results revealed that DNAss exhibited no significant association with the risk score (R = 0.1, p = 0.056). In contrast, RNAss displayed a positive correlation with the risk score (R = 0.38, p = 7e−14), indicating elevated tumor stemness in the high-risk group and a reduced degree of tumor cell differentiation (Figs. 11A and 11B).

Figure 11 Cancer stem cell analysis and single-cell analysis of the prognostic model.

(A and B) Cancer stem cell analysis in high-and low-risk groups in the TCGA cohort. (C–H) Single-cell analysis of six signature genes in HCC tissues.

Single-cell analysis of signature genes

The findings indicated that HCC tissues were primarily composed of malignant cells, with fibroblasts as the secondary component. FTH1, G6PD, PML, SLC1A5, SLC7A11, and STMN1 exhibited their highest expression levels in macrophages, B cells (in three cases), hepatic progenitor cells, and endothelial cells (Figs. 11C–11H). In summary, the distribution of the six signature genes of the model within HCC tissues displayed variations.

Validation of signature genes in databases

Immunohistochemical data for both liver normal tissue and tumor tissue was obtained from the HPA database. While accurately discerning differences in gene expression between liver normal and tumor tissues might be challenging, preliminary findings revealed elevated expressions of G6PD, STMN1, and PML in HCC tissues in comparison to normal liver tissues. Notably, protein expression data for SLC7A11 was unavailable. However, there was no significant disparity in the expression of FTH1 and SLC1A5 (Fig. 12A). Moreover, we observed variations in the expression levels of the six signature genes across different HCC cell lines (Fig. 12B). In summary, the expression of three out of the six signature genes exhibited significant differences between HCC tissues and normal liver tissues in the HPA database.

Figure 12 Validation of six signature genes in databases.

(A) Validation of protein expression of six signature genes in the HPA database (protein expression data for SLC7A11 was unavailable). (B) The expression of six signature genes in different HCC cell lines.

Validation of signature genes in protein level

Immunohistochemical analysis of 24 pairs of HCC tissues and their respective adjacent tissues revealed that there were no statistically significant differences in the expression of FTH1 and SLC1A5 between the tumor tissues and adjacent counterparts (Figs. 13A and 13B, Figs. S5A and S5B). However, the staining intensity of the signature proteins—PML, G6PD, SLC7A11, and STMN1—in the tumor tissues was notably stronger than that in the paired adjacent tissues (Figs. 13C–13F, Figs. S5C–S5F).

Figure 13 Validation of six signature genes in clinical tissue samples.

(A–F) Immunohistochemical staining of FTH1, SLC1A5, PML, G6PD, SLC7A11, and STMN1 in liver cancer tissue (left) and paired adjacent tissue (right), respectively.

Discussion

Primary liver cancer continues to pose a significant challenge with one of the lowest 5-year survival rates (21%) among malignancies. HCC predominates in the pathological classification of liver cancer. Early-stage HCC often presents with inconspicuous clinical symptoms, leading to diagnoses at an advanced stage in 50% of patients. While the emergence of molecularly targeted drugs and immunotherapy has provided a newfound advantage for advanced HCC patients, the scope of survival benefits remains constrained. Consequently, the imperative to explore novel prognostic biomarkers and therapeutic approaches has intensified. In addition to the tumor growth inhibition observed with Sorafenib in HCC, targeting ferroptosis has demonstrated its capacity to induce cancer cell death and impede tumor growth across various malignancies. Modulating the expression of cysteine desulfurase (NFS1) to induce ferroptosis has shown potential in curtailing the progression of lung adenocarcinoma (Alvarez et al., 2017). Erastin, an inducer of ferroptosis, heightens the sensitivity of lung cancer cells to cisplatin, while bolstering Erastin’s presence inhibits ovarian cancer invasion (Guo et al., 2018; Basuli et al., 2017). The combined application of sulfadiazine (SSZ) and piperine (PL) significantly elevates ROS levels, thereby inducing ferroptosis in pancreatic cancer cells (Yamaguchi, Kasukabe & Kumakura, 2018). Exploring a recently uncovered form of cell death, disulfidptosis, remains an uncharted frontier in various malignancies. As a response, our study has developed a prognostic model for HCC, anchored in ferroptosis-related genes linked to disulfidptosis.

We conducted an analysis of RNA-seq data obtained from HCC samples sourced from the TCGA, GEO, and ICGC databases. Subsequently, an unsupervised cluster analysis was applied to these HCC samples based on the transcriptome expression levels of 16 DE-DRG-FRGs associated with OS. Our investigation aimed to delineate disparities in survival probability and immune cell infiltration between the resulting clusters. The findings revealed that cluster B exhibited improved OS outcomes compared to cluster A. Moreover, cluster A demonstrated higher levels of immune cell infiltration, immune score and ESTIMATE score. Notably, cluster A displayed heightened infiltration of immune cells such as myeloid-derived suppressor cells (MDSCs), macrophages, and regulatory T cells. MDSCs (Gabrilovich, 2017), tumor-associated macrophage (TAM) (Cassetta & Pollard, 2018) and Treg cells (Tanaka & Sakaguchi, 2017) are recognized for their roles in promoting immunosuppression during tumorigenesis and disease progression. Consequently, it was conjectured that the shorter survival duration observed in cluster A might be attributed to its more pronounced immunosuppressive TME. The composition of the TME holds a pivotal role as a mediator affecting immune escape, tumor growth, invasion, and resistance to immunotherapy (Bagaev et al., 2021). An inhibitory TME has been associated with unfavorable prognoses in cancer patients.

A risk prognostic model was subsequently developed using LASSO and Cox regression analysis, based on six DRG-FRGs: FTH1, G6PD, PML, SLC1A5, SLC7A11, and STMN1. FTH1, responsible for encoding ferritin heavy subunits, emerged as a pivotal mediator of ferroptosis in bladder cancer cells induced by baicalin. Moreover, overexpressing FTH1 was observed to attenuate the anticancer effects of baicalin in both in vitro and in vivo settings (Kong et al., 2021). G6PD primarily functions in generating NADPH, a crucial electron donor for biosynthesis reactions, and is closely intertwined with cell growth and death processes (Yang et al., 2019). Notably, G6PD exhibits abnormal elevation in various cancer types, with its aberrant activation associated with the proliferation and invasion of several malignant tumor types (Yang, Stern & Chiu, 2021). PML expression is contingent upon the cell cycle and possesses the ability to activate TP53 or Rb/E2F pathways, thereby inhibiting apoptosis (Martin-Martin et al., 2016). Liying Han and colleagues reported a significant increase in SLC1A5 expression in glioblastoma tissues compared to low-grade gliomas (Han et al., 2022). Notably, the knockdown of SLC1A5 significantly curtailed the proliferation and invasion of glioma cells. Furthermore, the targeting of SLC1A5 substantially impeded the growth and proliferation of lung cancer cells (Hassanein et al., 2013). Previous studies have documented the overexpression of SLC7A11 in various cancer types, with a demonstrated role in tumor promotion. Cancer cells can augment tumor growth by upregulating SLC7A11 expression and inhibiting ferroptosis through diverse mechanisms (Koppula, Zhuang & Gan, 2021). SLC7A11 is also a disulfidptosis-related gene. Elevated SLC7A11 expression introduces vulnerabilities to cancer cells, rendering them notably sensitive to glucose or glutamine deficiencies. Consequently, it becomes easier to trigger disulfidptosis during glucose deprivation (Liu et al., 2023; Koppula, Zhuang & Gan, 2021; Lee & Roh, 2022). STMN1, an unstable phosphorylated microtubule-associated protein, is recognized as an oncogene. Its heightened expression is closely linked to unfavorable prognoses in multiple cancer types. The downregulation of STMN1 has been observed to impede the growth of gallbladder cancer cells and mitigate the Warburg effect (Wang et al., 2021). Elevated STMN1 expression in lung squamous cell carcinoma correlates with vascular invasion, decreased sensitivity to paclitaxel, and unfavorable prognostic outcomes (Bao et al., 2017). In a similar vein, Yaming Li and colleagues demonstrated that targeting STMN1 can mitigate chemotherapy resistance and metastasis in triple-negative breast cancer (Li et al., 2018). Our study affirms the efficacy of the risk prognostic model, constructed based on these six genes, in accurately predicting the survival of patients with HCC.

KEGG enrichment analysis revealed that genes in the high-risk group were predominantly associated with pathways involving the cell cycle, cytokine and cytokine receptor interaction, as well as DNA replication. Cell cycle dysregulation is one of the mechanisms underlying tumorigenesis and is intricately linked to metabolic reconfiguration and immune evasion (Liu, Peng & Wei, 2022). The division of tumor cells involves a mutation that enables the cell cycle to advance, inhibiting withdrawal. Numerous malignant tumors increasingly depend on residual cell cycle control mechanisms to avert genomically unstable reproduction. Thus, targeting cancer’s reliance on cell cycle control pathways has emerged as a promising therapeutic strategy (Matthews, Bertoli & de Bruin, 2022). Many studies have focused on the role of cytokines and cytokine receptors as potential targets for tumor intervention, including the inhibition of the tumor-promoting functions of IL-1β, IL-6, and TNF. Various antagonists targeting inflammatory cytokines or their receptors have been employed in patients with advanced cancer, potentially leading to short-term disease stabilization (Propper & Balkwill, 2022). In the cell cycle, a dividing cell must accurately replicate its DNA once to preserve genome integrity. Disruptions in this process can readily lead to the onset of diseases like cancer (Boyer, Walter & Sorensen, 2016). Tumors are characterized by sustained proliferation, evasion of apoptosis, and genomic instability (Macheret & Halazonetis, 2015). DNA replication stress could drive cancer progression and is considered a hallmark of the disease. Overall, the poorer prognosis observed in the high-risk group may be attributed to disturbances in the cell cycle, the production of tumor-promoting cytokines, and unstable DNA replication.

To the best of our knowledge, this is the inaugural investigation into the potential utilization of ferroptosis genes linked to disulfidptosis in the prognosis and tumor immunity of Hepatocellular Carcinoma. Nevertheless, our study is not without limitations. Firstly, the restricted number of disulfidptosis-related genes posed a limitation. Our prognostic model was constructed using ferroptosis-related genes linked to disulfidptosis, which may limit the specificity of our description of the correlation between disulfidptosis-related genes and the prognosis of HCC. To address this, further exploration necessitates the inclusion of additional disulfidptosis-related genes, large-scale clinical trials, and validation of molecular functions and mechanisms through in vitro and in vivo studies. Secondly, despite our validation involving two cohorts, substantial heterogeneity persisted among tumor samples characterized by diverse regional and ethnic attributes, and even within the individual samples. Nevertheless, notwithstanding these limitations, it is indisputable that our model holds clinical significance in predicting the tumor microenvironment and the survival of patients with HCC.

Supplemental Information

Supplemental Information 1 Supplementary material.

Click here for additional data file.

Additional Information and Declarations

Competing Interests

Author Contributions

Data Availability

The authors declare that they have no competing interests.

Jiayan Wei conceived and designed the experiments, performed the experiments, analyzed the data, prepared figures and/or tables, authored or reviewed drafts of the article, and approved the final draft.

Jinsong Wang conceived and designed the experiments, performed the experiments, analyzed the data, prepared figures and/or tables, authored or reviewed drafts of the article, and approved the final draft.

Xinyi Chen performed the experiments, prepared figures and/or tables, authored or reviewed drafts of the article, and approved the final draft.

Li Zhang conceived and designed the experiments, authored or reviewed drafts of the article, and approved the final draft.

Min Peng conceived and designed the experiments, authored or reviewed drafts of the article, and approved the final draft.

The following information was supplied regarding data availability:

The data is available at figshare and the following sites: 魏, 家燕 (2024). Ferroptosis-Related Genes Risk Model Associated with Disulfidptosis in Hepatocellular Carcinoma Prognosis. figshare. Dataset. https://doi.org/10.6084/m9.figshare.24971169.v1.

TCGA: TCGA-LIHC, https://www.cancer.gov/ccg/research/genome-sequencing/tcga;

GEO: GSE14520;

ICGC: LIRI-JP, https://dcc.icgc.org;

FerrDb: Driver+Suppressor+Marker genes:259, http://www.zhounan.org/ferrdb/legacy/operations/download.html;

GDSC: GDSC2-LIHC, https://www.cancerrxgene.org;

UCSC Xena: TCGA Liver Cancer (LIHC), http://xena.ucsc.edu;

STRING: http://string.embl.de;

TIMER 2.0: http://timer.cistrome.org;

CIBERSORT: https://cibersortx.stanford.edu;

cBioPortal: http://www.cbioportal.org;

MSigDB: https://www.gsea-msigdb.org/gsea/msigdb;

TISCH: http://tisch.comp-genomics.org;

HPA: https://www.proteinatlas.org;

CCLE: http://www.broadinstitute.org/ccle.

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
