# Peer review of "Novel application of the ferroptosis-related genes risk model associated with disulfidptosis in hepatocellular carcinoma prognosis and immune infiltration"

_PeerJ, doi:10.7717/peerj.16819_

## Round 0.1 · original submission · Major Revisions

Two reviewers gave specific suggestions for modification. Please revise the manuscript carefully.

**Language Note:** The review process has identified that the English language must be improved. PeerJ can provide language editing services - please contact us at [email protected] for pricing (be sure to provide your manuscript number and title). Alternatively, you should make your own arrangements to improve the language quality and provide details in your response letter. – PeerJ Staff

·

Basic reporting

The research have potential implications for Hepatocellular carcinoma (HCC) stratification and clinical trial.

Experimental design

1. Why only obtain 50 normal samples from the TCGA database for clinical information, survival data, gene mutation information, and RNA sequencing data (RNA-seq)?
2. Why only conduct single-cell analysis of 6 signature genes?
3. Why select FTH1, SLC1A5, PML, G6PD, SLC7A11, and STMN1 for immunohistochemical staining?

Validity of the findings

1. There are 71 up-regulated genes and only one down-regulated gene in tumor tissue. Does it because the source of tumor tissue?
2. For drug sensitivity analysis, the authors should perform analysis on IC50 in more than two risk groups.
3. In Fig.3A, why only conduct unsupervised clustering of 16 DE-DRG-FRGs?
4. What’s the significance for gene mutation and tumor mutation burden (TMB) analysis?

Reviewer 2 ·

Basic reporting

Please see Additional Comments.

Experimental design

Please see Additional Comments.

Validity of the findings

Please see Additional Comments.

Additional comments

The authors conducted a series of bioinformatics analyses to explore the role of some genes related to ferroptosis & disulfidptosis in HCC patient prognosis. The current study would be much improved if the authors address the following concerns:


------[Major Concerns about FIGURES, METHODS, RESULTS, and/or CONCLUSIONS]
1. In all FIGURES, it would be clear and more readable to BOTH provide figures with high resolution AND expand on figure legends by explaining the meanings of colors, groups, lines, and abbreviations. These revisions would greatly help readers to understand the results and their implications easily and efficiently. For example,
1.1 In all FIGURES' bar graphs, it would be more informative to display individual data points; in other words, please replace bar graphs by EITHER scatter plots with bars OR scatter plots (a pattern like PMID: 34537192, PMID: 37046252, and PMID: 37452367). Bar graphs have been shown to be misleading, because they cannot reveal variation/dispersion within data; instead, scatter plots with bars could be acceptable and scatter plots would be preferable (as confirmed by PMID: 25901488 and PMID: 28974579).
1.2 In all FIGURES' legends, it would be more rigorous to mention BOTH the sample size (the number of data points OR how many samples/patients were included) AND whether the data points were technical or biological replicates.
1.3 In all FIGURES' legends, it would be more rigorous to mention how the authors reported the data error (variation/dispersion): standard deviation (SD), confidence intervals (CI), or standard error of the mean (SEM, which would be not preferable).
1.4 The legend of Figure 9 seems oversimplified. Please mention more details.

2. In TITLE, it would be more informative to mention both "ferroptosis" and "disulfidptosis", because both genes were analyzed in this study.

3. In ABSTRACT:
3.1 It would be clearer to rewrite "The results demonstrated that the prognosis and immune features of the two clusters were significantly different" by mentioning what are "the two clusters". Likewise, please explain what are the "three cohorts" in "The RS played an independent prognostic role across all three cohorts.
3.2 It would be clearer to rewrite "Median RS divided HCC samples into high- and low-risk groups" by mentioning what was included in the "HCC samples" (that is, were these from TCGA, GEO, and/or ICGC databases?)
3.3 It would be more informative to rewrite "Further analysis involving cancer stem cell inûltration and drug sensitivity revealed notable differences in the two risk groups" by mentioning what were the "notable differences".
3.4 It would be clearer (easier to understand) to rewrite "Broadly speaking, the risk prognostic model constructed based on ferroptosis genes associated with disulfidptosis could well predict the survival time of HCC patients" by explaining what were " ferroptosis genes associated with disulfidptosis", which is a concept not introduced in sentences before this one.

4. In INTRODUCTION,
4.1 It seems unclear why the authors simultaneously analyzed ferroptosis- and disulfidptosis-related genes. Specifically, the authors defined "the ferroptosis-related gene (FRGs), which correlate with disulfidptosis-related genes (DRGs)" (in Paragraph 4), but the authors did not seem to provide enough rationale for what evidence (from literature) suggested that ferroptosis-related gene "correlate with" disulfidptosis-related genes.

5. In METHODS:
5.1 In 2.1 Data Acquisition ("The external validation cohorts GSE14520 and ICGC for HCC were then downloaded from the GEO and ICGC databases, respectively"), it would be more informative and rigorous to mention how many control and/or tumor samples were included in "cohorts GSE14520 and ICGC for HCC".

6. In RESULTS:
6.1 It would be clearer to end each paragraph in RESULTS with one sentence: "Together, these results suggest that ..." (a pattern like PMID: 37452367, PMID: 34715879, PMID: 34384362, PMID: 35965679, and PMID: 34537192), summarizing a paragraph AND highlighting the implications of all results in the paragraph.


------[Minor Concerns about writing]
1. Throughout the manuscript, it seems better to use Grammarly (https://www.grammarly.com/) to check & correct potential grammatical errors or typos. For example,
1.1 In Paragraph 2 of INTRODUCTION, it seems better to change "disulfdptosis" into "disulfidptosis".

2. In ABSTRACT:
2.1 It seems better to change "However, the identification of effective prognostic biomarkers for this population remains inadequate" into "However, the identification of effective prognostic biomarkers for HCC patients remains inadequate". After this revision, the sentence would be clearer, because the former did not mention who is "this population".

3. In INTRODUCTION:
3.1 In Paragraph 3 ("Studies have found that inducing ferroptosis in mice could not inhibit the occurrence of HCC but instead induce immunosuppression. However, the addition of immune checkpoint inhibitors (ICIs) and blocking Myeloidderived suppressor cells (MDSCs) could improve the survival of mice"), it would be clearer (easier to understand) to mention why the authors mentioned these two sentences. In other words, please point out what is meant by these two sentences (perhaps they suggested the importance of immune response in HCC?).
3.2 In Paragraph 4 ("Therefore, in this study, we integrated multiple DRGs- FRGs through high-throughput biomarker sequencing and constructed a prognostic risk model in HCC"), it seems better to change this sentence into "Therefore, in this study, we integrated multiple DRGs- FRGs through analyzing public sequencing datasets and constructed a prognostic risk model in HCC". After this revision, the sentence would be clearer and more rigorous, because the "high-throughput biomarker sequencing" was not done by the authors.

4. In METHODS:
4.1 In 2.1 Data Acquisition ("The correlation analysis between DRGs and FRGs was carried out, and 174 DRG-FRGs were screened out"), it seems better to change this sentence into "... 174 DRG-FRGs were picked out." After this revision, the sentence would be clearer, because "screen out" means removing something.

---

## Round 0.2 · Minor Revisions

As per the comments from R2, further revisions are still needed by the authors.

·

Basic reporting

The article shared professional structure, figures, tables and raw datas.

Experimental design

Research questions are well defined, relevant and meaningful.

Validity of the findings

All the datas are robust, statistically sound and controlled.

Reviewer 2 ·

Basic reporting

Please see Additional Comments.

Experimental design

Please see Additional Comments.

Validity of the findings

Please see Additional Comments.

Additional comments

Thank the authors for responding to the comments. However, the authors did not seem to address some issues thoroughly:

1. As to the previous comment 1.2 (1.2 In all FIGURES' legends, it would be more rigorous to mention BOTH the sample size (the number of data points OR how many samples/patients were included) AND whether the data points were technical or biological replicates.), the authors did not mention the sample size because they argued that "The production of the graphs was based on TCGA, GEO, and ICGC databases or online websites". However, these graphs are also based on samples, so it would be more informative and rigorous to touch on the sample size in these figures. For example, in the legend of Figure 2, please mention how many "Normal" and "Tumor" samples were used to derive the heatmap, what p-value & fold-change thresholds were used to identify the "differentially expressed DRG-FRGs", and how many "differentially expressed DRG-FRGs" (including both up- and down-regulated, respectively) were selected for the analyses in Figures 2C–G. Thus, please mention these details about the sample size in as many figures as possible.

2. As to the previous comment 3.4 (3.4 It would be clearer (easier to understand) to rewrite "Broadly speaking, the risk prognostic model constructed based on ferroptosis genes associated with disulfidptosis could well predict the survival time of HCC patients" by explaining what were "ferroptosis genes associated with disulfidptosis", which is a concept not introduced in sentences before this one.), the authors did mention the concept "ferroptosis genes associated with disulfidptosis" but did not define it. It would be more informative and clearer (easier to understand) to explain what are "ferroptosis genes associated with disulfidptosis" and how they were identified.

3. As to the previous comment 5.1 (5.1 In 2.1 Data Acquisition ("The external validation cohorts GSE14520 and ICGC for HCC were then downloaded from the GEO and ICGC databases, respectively"), it would be more informative and rigorous to mention how many control and/or tumor samples were included in "cohorts GSE14520 and ICGC for HCC".), the authors did respond to the comment, but it would be more informative to incorporate the reply into the manuscript's Methods (2.1 Data Acquisition).

---

## Round 0.3 · accepted · Accept

Two reviewers have agreed to accept this manuscript for publication. I agree with their decision.

Reviewer 2 ·

Basic reporting

Please see Additional Comments.

Experimental design

Please see Additional Comments.

Validity of the findings

Please see Additional Comments.

Additional comments

Thank the authors for responding to all of the comments. The current version has been improved.